# Use of family history taking for hereditary neoplastic syndromes screening in primary health care: A systematic review protocol

**Raphael Manhães Pessanha**[1☯], **Sara Isabel Pimentel de Carvalho Schuab**[2], **Karolini Zuqui Nunes**[2], **Luís Carlos Lopes-Júnior**[1☯]*

**1** Graduate Program in Public Health-PPGSC, Health Sciences Center at the Federal University of Espírito Santo (UFES), Vitoria, ES, Brazil, **2** Health Sciences Center at the Federal University of Espírito Santo (UFES), Vitoria, ES, Brazil

☯ These authors contributed equally to this work.
* lopesjr.lc@gmail.com

**Data Availability Statement:** No datasets were generated or analysed during the current study. All relevant data from this study will be made available upon study completion.

## Abstract

### Background

Although most neoplasms result from complex interactions between the individual's genome and the environment, a percentage of cases is particularly due to inherited alterations that confer a greater predisposition to the development of tumors. Hereditary Neoplastic Syndromes have a high psychosocial and economic burden, in addition to being characterized by an increased risk for one or more types of cancer, onset of malignancy at a young age, high risk of secondary malignancies, and occurrence in successive generations of the family. Personal and family history, as well as pedigree, may be useful resources to estimate the risk for developing cancer, especially in Primary Health Care settings.

### Objective

To identify and evaluate the evidence regarding the impact of using family history as a genomic technology for Hereditary Neoplastic Syndromes screening at Primary Health Care.

### Methods

This systematic review protocol was elaborated in compliance with the Preferred Reporting Items for Systematic Review and Meta-analysis Protocols (PRISMA-P). We will include all observational as well as experimental study designs published up to end of July 2022, and studies covering the impact of family history on screening for Hereditary Neoplastic Syndromes in primary health care. Qualitative studies, as well as guidelines, reviews, and studies undertaken in hospitals, outpatient clinic, or medical environments will be excluded. Five databases will be searched, including MEDLINE/PubMed, Cochrane Library, EMBASE, Web of Science, and LILACS. Additional sources will also be retrieved, including Clinical trials.gov-NIH, The British Library, and Google Scholar. No restriction to language or date will be employed in the search strategy. Three investigators will select studies independently, perform data extraction, and conduct the critical appraisal of the risk of bias and overall

**Funding:** The author(s) received no specific funding for this work.

**Competing interests:** The authors have declared that no competing interests exist.

quality of the selected studies according to their designs. Regarding data synthesis, the study characteristics will be summarized and presented in tables and results will be presented according to the study design. A qualitative synthesis of data will also be provided in this systematic review.

## Discussion

To the best of our knowledge, this systematic review will be the first to identify and critically assess evidence regarding the impact of using family history as a genomic technology for Hereditary Neoplastic Syndromes screening in Primary Health Care settings worldwide. This study is expected to provide consistent evidence that will aid the primary care decision-makers regarding hereditary cancer screening, thus helping individuals and families at risk for cancer.

### PROSPERO registration number

CRD42020166211.

## 1. Introduction

Although most neoplasms result from complex interactions between the individual's genome and the environment, a percentage of cases is particularly due to inherited alterations that confer a greater predisposition to the development of tumors. About 5 to 10% of all cancers are associated with hereditary predisposition [1].

Hereditary predisposition to cancers exhibits an autosomal dominant pattern of inheritance and occurs in individuals with germline mutations that often have high penetrance, making them more susceptible to developing certain Hereditary Neoplastic Syndromes (HNS) [2]. The genes associated with HNS tend to belong to the group of tumor suppressor genes, which, when altered, do not adequately play their regulatory role in cell growth, or to the group of DNA repair genes. Such HNSs can also be associated with oncogenes when inappropriately activated [3].

Indeed, in the last two decades, knowledge about the molecular mechanisms underlying the origin of cancer has advanced substantially, which allowed identifying several genes involved in the development of cancer. Such knowledge culminated in the identification of genes associated with specific cancer predisposition syndromes. More than 50 HNSs have already been defined so far [4, 5]. These discoveries supported the emergence of molecular tests for the diagnosis of HNS, among other types of cancer, and stimulated the development of clinical assessment programs, as well as genetic counseling for individuals and families at risk [6].

Hereditary Neoplastic Syndromes place a high psychosocial and economic burden, besides posing an increased risk of one or more types of cancer, onset of malignancy at a young age, secondary malignancies, and occurrence in successive generations of the family [1, 5], as well as of developing several forms of cancer throughout life (approximately 80% for the most prevalent tumors in the syndromes). Thus, identifying family members at risk is essential for early diagnosis and preventive care [6], to reducing cancer-related morbidity and mortality, as well as costs for health systems [6, 7].

Among the items comprising the assessment, family history of cancer is the one deserving the greatest highlight, since the identification of many HNSs occurs through a properly taken

and validated history [5, 8, 9]. When taking and interpreting family history data, health professionals shall be aware of some clinical markers that indicate inherited susceptibility to cancer, referred to in the literature as "red flags", which include: (a) young age at diagnosis (in general under 50 years); (b) bilateral tumors in paired organs; (c) presence of multiple distinct tumors in the same organ; (d) multiple primary tumors in the same individual; (e) presence of tumors in two or more first- or second-degree relatives; (f) "constellation" of tumors in a subject or his/her family, recognized as part of an HNS already described in the literature; and (g) associations of cancer with benign lesions [5, 8].

The collection of personal and family information for the purpose of genetic screening takes place through interviews and/or questionnaires. The family history collected may be different and depends on the different methods used, which may affect its predictive value. It should be noted that the analytical validity of the information collected, in terms of sensitivity and specificity, depends on the self-report of family members regarding the family history of cancer in their relatives (as to the degree of kinship), types of tumors, and age of the patient at diagnosis [10, 11]. However, this shall be confirmed through medical records, such as results of anatomopathological exams, death certificate, or histopathology of the tumors referred to in the family [8]. Furthermore, the consistency of the self-reported family history has implications for risk assessment, as well as for the management and definition of clinical diagnostic criteria for the various HNSs [12, 13].

Personal and family history, as well as pedigree, may be useful resources to estimate the risk for developing cancer, especially in PHC settings. Moreover, the knowledge of the degree of risk is important when deciding whether to perform a genetic testing, indicate clinical screening procedures, and use chemoprevention measures [14]. Several methods and statistical models have been proposed to quantify the risk associated with a family history of cancer [8]. Once the diagnosis is made and the risk of developing cancer is determined, according to the clinical characteristics of the family, the health team guides the proband on the indication of predictive genetic tests, on their individualized follow-up and that of their families, as well as on the specific prevention and screening program if necessary [8, 15].

Heredity plays a fundamental role in the etiology of cancer, and this needs to be assessed at primary health care. Furthermore, the survey of family history, as well as the use of genetic tests, allow the identification of a significant portion of individuals who are at high risk of developing cancer. Hence, the purpose of this study is to identify and evaluate the evidence regarding the impact of using family history as a genomic technology for Hereditary Neoplastic Syndromes screening at Primary Health Care.

## 2. Methods

This systematic review protocol has been created and has high compliance with the *Preferred Reporting Items for Systematic Reviews and Meta-Analyses Protocols*-PRISMA-P [16]. Moreover, this protocol is registered with PROSPERO/UK (registration ID: CRD42020166211).

### Search strategy

The search strategy will be performed in five electronic databases: *Medical Literature Analysis and Retrieval System Online–MEDLINE/PubMed, Cochrane Library, EMBASE, Web of Science* and *Latin America and the Caribbean Literature on Health and Science–LILACS*; all search strategies will consider records from inception in each database up to end of July 2022. Additional sources will be also searched, including Clinical trials.gov-NIH, The British Library, and Google Scholar. No restriction to language or date will be employed in the search strategy. Furthermore, we will also scrutinize the reference lists of articles searched for additional studies

[17]. The PECO (Population/Exposure/Comparison/Outcomes) acronym [18] was used to elaborate our research question, considering P = (People with Hereditary Neoplastic Syndrome users of Primary Health Care), E = (Family history), C = (Not applicable), O = (Screening for hereditary cancer). Hence, our research question was: *"What scientific evidence is available on the impact of using family history as a genomic technology for screening for hereditary neoplastic syndromes in Primary Health Care"*?

The EndNote™ will be used to organize and manage all the studies retrieved. Study selection will be carried out for three independently researchers (RMP, KZN and LCLJ) using Rayyan™ application. First, we will screen for controlled descriptors, for instance, MeSH terms, DeCs, and their synonyms, and keywords will be identified. The Boolean operators "AND" and "OR" will be employed to combine the descriptors [19, 20]. The preliminary pilot search strategy combining MeSH terms, synonyms, and keywords that will be used in MEDLINE/PubMed is depicted in the Table 1.

## Eligibility criteria

We will include all observational and experimental study designs, published up to end of July 2022, and studies covering the impact of family history on screening for HNS in primary health care. Qualitative studies, guidelines, reviews, and studies undertaken in hospitals,

**Table 1. Preliminary pilot search strategy in MEDLINE/PubMed.**

| Database | Search strategy |
|---|---|
| MEDLINE/ PubMed | **#1** (Primary Health Care [MeSH Terms] OR Care, Primary Health [Title/Abstract] OR Health Care, Primary [Title/Abstract] OR Primary Healthcare [Title/Abstract] OR Care, Primary [Title/Abstract] OR Public Health [MeSH Terms] OR Health, Public [Title/Abstract] OR Community Health [Title/Abstract] OR Health, Community [Title/Abstract] OR Environment, Preventive Medicine and Public Health [Title/Abstract]) |
| | **#2** (Neoplastic Syndromes, Hereditary [MeSH Terms] OR Cancer Syndromes, Hereditary [Title/Abstract] OR Hereditary Neoplastic Syndromes [Title/Abstract] OR Hereditary Neoplastic Syndrome [Title/Abstract] OR Neoplastic Syndrome, Hereditary [Title/Abstract] OR Syndrome, Hereditary Neoplastic [Title/Abstract] OR Syndromes, Hereditary Neoplastic [Title/Abstract] OR Hereditary Cancer Syndromes [Title/Abstract] OR Cancer Syndrome, Hereditary [Title/Abstract] OR Hereditary Cancer Syndrome [Title/Abstract] OR Syndrome, Hereditary Cancer [Title/Abstract] OR Syndromes, Hereditary Cancer [Title/Abstract] OR Genetic Predisposition to Disease [MeSH Terms] OR Genetic Susceptibility [Title/Abstract] OR Genetic Susceptibilities [Title/Abstract] OR Susceptibilities, Genetic [Title/Abstract] OR Susceptibility, Genetic [Title/Abstract] OR Genetic Predisposition [Title/Abstract] OR Genetic Predispositions [Title/Abstract] OR Predispositions, Genetic [Title/Abstract] OR Predisposition, Genetic [Title/Abstract]) |
| | **#3** #1 AND #2 |
| | **#4** (Medical history taking [MeSH Terms] OR History Taking, Medical [Title/Abstract] OR Family Medical History [Title/Abstract] OR Family Medical Histories [Title/Abstract] OR Medical Histories, Family [Title/Abstract] OR Medical History, Family [Title/Abstract] OR Family Health History [Title/Abstract] OR Family Health Histories [Title/Abstract] OR Health Histories, Family [Title/Abstract] OR Health History, Family [Title/Abstract] OR Family History, Medical [Title/Abstract] OR Family Histories, Medical [Title/Abstract] OR Medical Family Histories [Title/Abstract] OR Medical Family History [Title/Abstract] OR Family History, Health [Title/Abstract] OR Family Histories, Health [Title/Abstract] OR Health Family Histories [Title/Abstract] OR Health Family History [Title/Abstract]) |
| | **#5:** (Early Detection of Cancer [MeSH Terms] OR Cancer Early Detection [Title/Abstract] OR Cancer Screening [Title/Abstract] OR Screening, Cancer [Title/Abstract] OR Cancer Screening Tests [Title/Abstract] OR Cancer Screening Test [Title/Abstract] OR Screening Test, Cancer [Title/Abstract] OR Screening Tests, Cancer [Title/Abstract] OR Test, Cancer Screening [Title/Abstract] OR Tests, Cancer Screening [Title/Abstract] OR Early Diagnosis of Cancer [Title/Abstract] OR Cancer Early Diagnosis [Title/Abstract]) |
| | **#6** #3 AND #4 AND #5 |

outpatient clinics, or medical environments will be excluded. Handsearching will be held in the reference lists to seeking additional studies. Moreover, no restriction to language or date will be employed in the search strategy. Regarding the language, the authors are fluent in English, Portuguese and Spanish. For articles selected in languages other than English, Portuguese and Spanish, the authors will count on the support of Letters Faculty as well as the Graduate Program in Linguistics at the University which are bonded for the translation of articles to be included in this review.

## Study selection

Initially, all the records scrutinized from the 5 electronic databases will be imported into End-Note™. Thus, the duplicate studies will be removed. Three independent researchers (RMP, KZN and LCLJ) will search and screen the records by titles and abstract into Rayyan™ app. After the initial screening, the full text of studies retrieved will be assessed for inclusion/exclusion by three independent researchers in order to minimize the bias using Rayyan™ app. Disagreements in selected studies will be figured out by discussion and consensus among the three reviewers. A flowchart will summarize the study selection process in line with the PRISMA 2020 statement [21] (Fig 1).

## Data extraction and data synthesis

Three reviewers (RMP, KZN and LCLJ) will perform data extraction for each included study based in forms previously published [17, 20, 22–24]. The expected completion date for this systematic review is November 30, 2022.

PRISMA 2020 flow diagram for new systematic reviews which included searches of databases, registers and other sources

**Fig 1. PRISMA flowchart [21].**

Information to be extracted includes, a) identification of the study and objectives; b) study population and baseline characteristics; c) type of exposure; d) study methodology; e) recruitment methods; f) times of measurement; g) follow-up; h) outcomes; i) main findings; j) clinical and epidemiological significance; and k) conclusions. Table 2 shows the standardized form elaborated for data extraction. It is stand out that data extraction will be independently cross-checked by the three reviewers (RMP, KZN and LCLJ).

### Critical appraisal of the studies included

Initially, the level of evidence will be identified based on the study design and according to the evidence hierarchy which classifies levels I and II as strong, III to V as moderate, and VI to VII as weak [25]. Thus, the internal validity and risk of bias of Randomized Controlled Trials will be assessed using the revised Cochrane Risk-Of-Bias tool for randomized trials (RoB 2) [26]. Otherwise, to assess Non-Randomized Controlled Trials, the Risk of Bias in Non-randomized Studies of Interventions (ROBINS-I) will be used [27]. In addition, Newcastle-Ottawa Scale (NOS) [28] will be used for evaluating the internal validity and risk of bias for cohort studies. For case-control studies, the Critical Appraisal Skills Programme (CASP) tool [29] will be used. Regarding the cross-sectional studies, we will assess using The Agency for Healthcare Research and Quality (AHRQ) tool [30]. The same three reviewers (RMP, KZN and LCLJ) will perform the critical appraisal in an independent manner.

Regarding data synthesis, the study characteristics will be summarized and presented in tables and results will be presented according to the study design, and a qualitative synthesis of the data in this systematic review will be provided.

**Assessment of publication bias.**   For assessing the publication bias, a funnel plot will be examined. Following the approach proposed by Duval and Tweedie [31] the number of studies that are missing from the funnel plot will be estimated, if any. The effect size after the imputation of these missing studies will be estimated by the trim-and-fill method [31] as well as by the Egger's test [32].

### Ethics issues and dissemination

No ethical approval is required for this study design. Moreover, the systematic review will be reported following the *Preferred Reporting Items for Systematic Reviews and Meta-Analyses—* PRISMA 2020 statement [21]. Regarding the plans of dissemination, we intend disclose the results via peer-reviewed publication and presentations in international conferences.

## 3. Discussion

Indeed, identifying individuals who are at risk for HNS is important for several reasons. First, because affected individuals have much higher cumulative vital risk than the general population for developing multiple neoplasms. Second, the relatives of an affected individual may be at risk, as most of these genetic diseases segregate into families following an autosomal dominant pattern of inheritance. Third, because intensive screening measures are effective in enabling earlier diagnoses. Fourth, because the identification of mutation carriers allows delineating strategies for risk reduction, chemoprevention, and prophylactic surgeries [8]. Thus, identifying family members at risk of development of HNS is essential for preventive care [6], with a view to reducing cancer-related morbidity and mortality, as well as costs for health systems [7, 15]. Furthermore, it is important to distinguish cases where the malignant neoplasm is sporadic, or family grouping, or hereditary [3, 4, 8].

In addition, early detection combined with the longitudinally of care in PHC can lead to a reduction in the number of patients diagnosed at an advanced stage of the disease, reflecting

**Table 2. Data extraction form based on previous publications [17, 20, 22–24].**

| *Study number*: | Level of evidence: Methodological Appraisal tool: |
|---|---|
| **STUDY CARACTERISTICS** | |
| Authors | |
| Title | |
| Year of publication | |
| Country | |
| Conflicts of interests | |
| Sponsorship | |
| Background | |
| Rationale | |
| Hypothesis tested | |
| Objectives | |
| *Methods* | |
| Methodology is reported according with STROBE (observational studies) or CONSORT (clinical trials) | |
| (    ) Yes | |
| (    ) No | |
| (    ) Partially | |
| Study design | |
| Local: | |
| Sample size and calculation: | |
| Inclusion criteria (definition of exposure of interest) | |
| Exclusion criteria | |
| Confounding factors/Interaction factors considered | |
| Ethical aspects | |
| Procedure for data collection: | |
| • Collection period: | |
| • Procedures: | |
| Instruments for data collection | |
| Outcomes / Evaluation of outcomes | |
| • Primary outcome: | |
| • Secondary outcome: | |
| Follow-up | |
| Statistical analysis | |
| If, cohort study | I. Number of participants in the exposed and unexposed cohort: |
| | II. Number of participants in each group: |
| | III. Comparability of exposed and unexposed cohorts |
| | IV. Contamination (unexposed patient being exposed): |
| | V. Follow-up period: |
| | VI. Dropouts: |
| If, case-control study | I. Criteria for selection of cases: |
| | II. Criteria for selection of controls: |
| | III. Comparability of groups: |
| | IV. Dropouts: |

(*Continued*)

**Table 2.** (Continued)

| Study number: | Level of evidence: Methodological Appraisal tool: |
|---|---|
| If, experimental or quase-experimental study | **a)** Trial Register: |
| | **b)** Trial arms: |
| | • Experimental Group: |
| | **c)** Randomization: |
| | **d)** Masking: |
| | **e)** Intervention protocol: |
| | **f)** Per-protocol and modified intention-to-treat analyses: |
| | • Per-protocol: |
| | • Intention-to-treat: |
| | • Dropouts: |
| *Results* | |
| Main results | |
| Clinical-Epidemiological Significance | |
| Limitations of the study | |
| Strengths of the study | |
| *Conclusions* | |
| Main conclusions | |
| Implication for clinical practice and research or for decision-makers / stakeholders | |

the minimization of costs for health systems [7, 8, 33–39]. The main potential limitations may be the predominance of cross-sectional studies that might limit the generalizability of the results, as well as the different instruments for taking family history that may not be psychometrically validated tools. Also, the different instruments used for collecting family history may hamper the collation of outcomes.

## 4. Conclusion

To the best of our knowledge, this systematic review will be the first to identify and critically assess evidence regarding the impact of using family history as a genomic technology for Hereditary Neoplastic Syndromes screening in Primary Health Care settings worldwide. It is expected that this study may provide consistent evidence that will aid primary care decision-makers with regards to hereditary cancer screening and therefore, to help individuals and families at risk for cancer.

## Supporting information

**S1 Checklist. PRISMA-P 2015 checklist.**
(DOCX)

## Acknowledgments

**Disclaimer:** The views of the authors do not necessarily reflect those of the NHS, NIHR or the Department of Health.

## Author Contributions

**Conceptualization:** Raphael Manhães Pessanha, Luís Carlos Lopes-Júnior.

**Data curation:** Raphael Manhães Pessanha, Karolini Zuqui Nunes, Luís Carlos Lopes-Júnior.

**Formal analysis:** Raphael Manhães Pessanha, Sara Isabel Pimentel de Carvalho Schuab, Karolini Zuqui Nunes, Luís Carlos Lopes-Júnior.

**Investigation:** Luís Carlos Lopes-Júnior.

**Methodology:** Luís Carlos Lopes-Júnior.

**Project administration:** Luís Carlos Lopes-Júnior.

**Software:** Luís Carlos Lopes-Júnior.

**Supervision:** Luís Carlos Lopes-Júnior.

**Validation:** Raphael Manhães Pessanha, Sara Isabel Pimentel de Carvalho Schuab, Karolini Zuqui Nunes, Luís Carlos Lopes-Júnior.

**Visualization:** Raphael Manhães Pessanha, Sara Isabel Pimentel de Carvalho Schuab, Karolini Zuqui Nunes, Luís Carlos Lopes-Júnior.

**Writing – original draft:** Raphael Manhães Pessanha, Sara Isabel Pimentel de Carvalho Schuab, Karolini Zuqui Nunes, Luís Carlos Lopes-Júnior.

**Writing – review & editing:** Raphael Manhães Pessanha, Karolini Zuqui Nunes, Luís Carlos Lopes-Júnior.

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
