## [Decision Letter · Decision Letter 0]

25 May 2022

PONE-D-21-25767

Title : Use of family history taking for Hereditary Neoplastic Syndromes screening in Primary Health Care: a systematic review protocol

PLOS ONE

Dear Dr. Lopes-Junior,

Thank you for submitting your manuscript to PLOS ONE. After careful consideration, we feel that it has merit but does not fully meet PLOS ONE’s publication criteria as it currently stands. Therefore, we invite you to submit a revised version of the manuscript that addresses the points raised during the review process.

The manuscript has been seen by a reviewer and their comments may be seen below.

The reviewer has provided some minor methodological suggestions to further improve the systematic review.  Could you please revise the manuscript to carefully address the concerns raised?

We look forward to receiving your revised manuscript.

Kind regards,

Lucinda Shen, MSc

Staff Editor

PLOS ONE

Journal Requirements:

Reviewers' comments:

Reviewer's Responses to Questions

**Comments to the Author**

1. Does the manuscript provide a valid rationale for the proposed study, with clearly identified and justified research questions?

Reviewer #1: Yes

2. Is the protocol technically sound and planned in a manner that will lead to a meaningful outcome and allow testing the stated hypotheses?

Reviewer #1: Yes

3. Is the methodology feasible and described in sufficient detail to allow the work to be replicable?

Reviewer #1: Yes

4. Have the authors described where all data underlying the findings will be made available when the study is complete?

Reviewer #1: Yes

5. Is the manuscript presented in an intelligible fashion and written in standard English?

Reviewer #1: Yes

6. Review Comments to the Author

You may also provide optional suggestions and comments to authors that they might find helpful in planning their study.

Reviewer #1: Main Theme of the manuscript:

In this manuscript, the authors wanted to identify and evaluate the evidence regarding the impact of family history as a genomic technology for Hereditary Neoplastic Syndromes (HNS) screening in Primary health care. My overall impression of the manuscript is written well. The personal and family history is a great resource to estimate the risk of developing cancer in the family members and it is a critical tool for the physicians to predict the possible outcomes and address health maintenance for the patients, especially in the primary health care settings.

Comments:

1. It is good to have no restriction on language for collecting the data from the published studies but how authors are going to deal with the data from the non-English origin?

2. Authors also need to address the potential publication and sample size bias.

3. At least three reviewers need to independently check each selected article to minimize the bias.

4. Data extraction should be independently cross-checked.

5. The different instruments used for collecting family history may hamper the collation of outcomes.

6. There are a few minor typos and grammatical errors. The author must check the grammatical errors thoroughly.

7. PLOS authors have the option to publish the peer review history of their article (what does this mean?). If published, this will include your full peer review and any attached files.

Reviewer #1: No

---

## [Author Response · Author response to Decision Letter 0]

29 May 2022

Response to Reviewers

Vitória, ES, Brazil, May 28th, 2022

Minor revision

PONE-D-21-25767

Title : Use of family history taking for Hereditary Neoplastic Syndromes screening in Primary Health Care: a systematic review protocol

Dear Dr. Lopes-Junior,

Thank you for submitting your manuscript to PLOS ONE. After careful consideration, we feel that it has merit but does not fully meet PLOS ONE’s publication criteria as it currently stands. Therefore, we invite you to submit a revised version of the manuscript that addresses the points raised during the review process.

The manuscript has been seen by a reviewer and their comments may be seen below.

The reviewer has provided some minor methodological suggestions to further improve the systematic review. Could you please revise the manuscript to carefully address the concerns raised?

Response: Dear Dr. Lucinda Shen

Thank you for the opportunity to review the manuscript after the reviewers' suggestions and recommendations.

All points were addressed and/or clarified in this new version. In addition, we responded item by item to the questions raised by the reviewers in this letter.

Reviewers' comments:

Reviewers' comments:

Reviewer's Responses to Questions

Comments to the Author

1. Does the manuscript provide a valid rationale for the proposed study, with clearly identified and justified research questions?

Reviewer #1: Yes

 2. Is the protocol technically sound and planned in a manner that will lead to a meaningful outcome and allow testing the stated hypotheses?

Reviewer #1: Yes

 3. Is the methodology feasible and described in sufficient detail to allow the work to be replicable?

Reviewer #1: Yes

 4. Have the authors described where all data underlying the findings will be made available when the study is complete?

Reviewer #1: Yes

 5. Is the manuscript presented in an intelligible fashion and written in standard English?

Reviewer #1: Yes

 6. Review Comments to the Author

You may also provide optional suggestions and comments to authors that they might find helpful in planning their study.

Reviewer #1: Main Theme of the manuscript:

In this manuscript, the authors wanted to identify and evaluate the evidence regarding the impact of family history as a genomic technology for Hereditary Neoplastic Syndromes (HNS) screening in Primary health care. My overall impression of the manuscript is written well. The personal and family history is a great resource to estimate the risk of developing cancer in the family members and it is a critical tool for the physicians to predict the possible outcomes and address health maintenance for the patients, especially in the primary health care settings.

Response: Thank you so much for your positive feedback and careful review.

Comments:

1. It is good to have no restriction on language for collecting the data from the published studies but how authors are going to deal with the data from the non-English origin?

Response: Regarding the language, the authors are fluent in English, Portuguese and Spanish. For articles selected in languages other than English, Portuguese and Spanish, the authors will count on the support of Letters Faculty as well as the Graduate Program in Linguistics at the University which are bonded for the translation of articles to be included in this review.

2. Authors also need to address the potential publication and sample size bias.

Response: We have added this issue in the paper. "Publication bias will be checked with funnel plots and Egger’s test. 

3. At least three reviewers need to independently check each selected article to minimize the bias.

Response: Right. We have added this information throughtout the text.

4. Data extraction should be independently cross-checked.

Response: We agree with you. All the steps of sthis systematic review will be independently cross-checked. We wrote this in the data extraction section of the paper for clarity, as suggested. Thanks!

5. The different instruments used for collecting family history may hamper the collation of outcomes.

Response: Indeed, this is an important point to be considered. Thanks for this timely comment. We have addressed this issue as a possible limitation of this review.

6. There are a few minor typos and grammatical errors. The author must check the grammatical errors thoroughly.

Response: The few minor typos and grammatical errors were addressed.

---

## [Decision Letter · Decision Letter 1]

28 Jun 2022

Title : Use of family history taking for Hereditary Neoplastic Syndromes screening in Primary Health Care: a systematic review protocol

PONE-D-21-25767R1

Dear Dr. Lopes-Junior,

We’re pleased to inform you that your manuscript has been judged scientifically suitable for publication and will be formally accepted for publication once it meets all outstanding technical requirements.

Kind regards,

Dylan A Mordaunt, MD, MPH, FRACP

Academic Editor

PLOS ONE

Additional Editor Comments (optional):

This study answers an interesting question. I think the identification of studies could be challenging and I'm not certain the study design will be sensitive enough to identify all possible papers, mainly because of the use of only one search database and a question of whether the search string used in PubMed is adequate. However, the authors have presumably identified some papers in their initial testing and the protocol has already been submitted to PROSPERO- changing the protocol in this manuscript would result in a measured reduction in quality based on formal quality assessment scaled like AMSTAR2, so the trade-off results in zero gain. I would also add grey literature searches and a search of reference lists from identified papers.

With regards to the criteria for publication:

1. The study presents the results of original research.

2. Results reported have not been published elsewhere.

3. Experiments, statistics, and other analyses are performed to a high technical standard and are described in sufficient detail.

4. Conclusions are presented in an appropriate fashion and are supported by the data.

5. The article is presented in an intelligible fashion and is written in standard English.

6. The research meets all applicable standards for the ethics of experimentation and research integrity.

7. The article adheres to appropriate reporting guidelines and community standards for data availability.

Reviewers' comments:

Reviewer's Responses to Questions

**Comments to the Author**

1. Does the manuscript provide a valid rationale for the proposed study, with clearly identified and justified research questions?

Reviewer #1: Yes

2. Is the protocol technically sound and planned in a manner that will lead to a meaningful outcome and allow testing the stated hypotheses?

Reviewer #1: Yes

3. Is the methodology feasible and described in sufficient detail to allow the work to be replicable?

Reviewer #1: Yes

4. Have the authors described where all data underlying the findings will be made available when the study is complete?

Reviewer #1: Yes

5. Is the manuscript presented in an intelligible fashion and written in standard English?

Reviewer #1: Yes

6. Review Comments to the Author

You may also provide optional suggestions and comments to authors that they might find helpful in planning their study.

Reviewer #1: The authors have addressed all the concerns raised by me. The the protocol technically sound and planned in a manner that will lead to a meaningful outcome in the future studies.

7. PLOS authors have the option to publish the peer review history of their article (what does this mean?). If published, this will include your full peer review and any attached files.

Reviewer #1: No

---

## [Editor Report · Acceptance letter]

30 Jun 2022

PONE-D-21-25767R1 

Use of family history taking for Hereditary Neoplastic Syndromes screening in Primary Health Care: a systematic review protocol 

Dear Dr. Lopes-Júnior:

I'm pleased to inform you that your manuscript has been deemed suitable for publication in PLOS ONE. Congratulations! Your manuscript is now with our production department. 

Kind regards, 

on behalf of

Associate Professor Dylan A Mordaunt 

Academic Editor

PLOS ONE